# Respiratory frequency-tunable dynamic imaging for lung function: New exam method using chest X-ray cine imaging considering various respiratory diseases

**Takehiko Abe**[1,2]*, **Norifumi Yoshida**[1,2], **Tetsuo Shimada**[3], **Masanao Nakashima**[4], **Atsushi Nagai**[4]

1 Department of Radiology, Teikyo University, Itabashi-ku, Tokyo, Japan, 2 Radwisp Pte. Ltd, The Central, Singapore, Singapore, 3 Faculty of Engineering, Sanjo City University, Sanjo, Niigata, Japan, 4 Department of Respiratory Medicine, Shin-Yurigaoka General Hospital, Asao-ku, Kawasaki, Kanagawa, Japan

* abe@radwisp.com

**Data Availability Statement:** Data is indicated by Supporting information (S9_date).

## Abstract

### Objectives

A convenient way to conduct pulmonary function tests while preventing infectious diseases was proposed, together with countermeasures for severe coronavirus disease 2019 (COVID-19). The correlation between diagnosis result and diagnosis result was examined for patients with mild chronic obstructive pulmonary disease (COPD) of the most abounding as a subject of spirometry, and the possibility of using this method as an alternative to spirometry was examined.

### Setting

This study was conducted in Kanagawa, Japan.

### Participants

Ten normal volunteers and 15 volunteers with mild COPD participated in this study.

### Outcome measures

All images were taken by EXAVISTA (Hitachi, Japan) between October 2019 and February 2020. Continuous fluoroscopic images were taken in 12.5 frames per second for 10–20 s per subject. Images that do not adopt the automatic image processing of the equipment and only carry out the signal correction of each pixel were used for the analysis.

### Results

The mean total dose for all volunteers was 0.2 mGy. There was no major discrepancy in the detection of lung field geometry, and no diagnostic problems were noted by the radiologist and physician.

**Funding:** The author(s) received no specific funding for this work.

**Competing interests:** TA and NY are the founders of the company concerning this technology which has "no" financial assistance.

## Conclusions

Existing X-ray cine imaging was used to extract frequency-tunable imaging. It is possible to identify abnormal regions on the images compared to spirometry, and it does not require maximum effort respiration; therefore, it is possible to perform a stable examination because the patient's physical condition and the ability of laboratory technicians on the day are less affected. This can also be used as a countermeasure in examining patients with infectious diseases.

## Trial registration

UMIN UMIN000043868.

## Introduction

Severe coronavirus disease 2019 (COVID-19) is a prevalent infectious disease worldwide, and healthcare organizations have to create more stringent infectious disease prevention systems than ever before [1–3]. In response to these circumstances, respiratory societies in different countries issued a documentation to stop performing spirometry, which delivers the patient's breath into the device, as much as possible to prevent infectious diseases [4, 5].

In addition to spirometry, scintigraphy is also used to evaluate pulmonary ventilation [6]. However, scintigraphy is difficult to handle due to the need for radiopharmaceuticals, and the patient burden of X-ray exposure is high. Computed tomography (CT) has also been used to assess pulmonary ventilation [7]. Although regional information, such as pneumonia and atelectasis, can be obtained in CT, the patient burden of X-ray exposure is big, similar to scintigraphy. Both scintigraphy and CT examinations are very expensive, and their financial burden to patients is substantial. There are also many hospitals that adopt an appointment system due to cost and long examination hours; however, there is a disadvantage that needed examinations are not possible in ambulatory treatment.

In this study, a convenient way to carry out pulmonary function tests while preventing infectious diseases was proposed, together with countermeasures for COVID-19.

A respiratory examination using continuous fluoroscopic images was proposed. Continuous fluoroscopy is a simple method with less exposure than scintigraphy and CT, and it does not require radiopharmaceuticals or contrast media. However, to obtain test results similar to those of spirometry, patients should also be forced to breathe strongly [8], and similar test methods should be avoided in terms of infectious disease prevention. Therefore, the proposed method does not perform forced expiration unlike spirometry.

This time, the correlation between diagnosis result and diagnosis result was examined for patients with mild COPD of the most abounding as a subject of spirometry [9], and the possibility of using this method as an alternative to spirometry was examined.

## Materials and methods

### 1. Data collection

This study was approved by Shin-yurigaoka General Hospital Ethics Committee, and all patients agreed to participate in the study. All images were taken by EXAVISTA (Hitachi, Japan) at Shin-yurigaoka General Hospital between October 2019 and February 2020.

Ten normal volunteers and 15 volunteers with mild COPD participated in this study. Seven and eight patients with COPD were ranked by GOLD1 and GOLD2, respectively. All

individuals with COPD were definitively diagnosed based on spirometry. The mean age of normal volunteers was $45.1 \pm 9.2$ years, while that of patients with GOLD1 and GOLD2 was $73.9 \pm 12.0$ years and $76.4 \pm 5.0$ years, respectively. Twenty-one males and four females were included. Written informed consent was obtained from all. This study was approved by the ethics committee of Shin-yurigaoka General Hospital (Approval No. 22–1).

Continuous fluoroscopic images in anteroposterior direction and in recumbent position were used for evaluation. Image areas were acquired using a 43 cm × 43 cm flat panel detector with a matrix number of 1024 × 1024 pixels and a density gradation of 4096. The distance from the X-ray focal point to the detector was 847 mm. The tube voltage and current were automatically determined by the fluoroscopy system before the inspection and fixed during the inspection. Continuous fluoroscopic images were taken in 12.5 frames per second for 10–20 s per subject. Images that do not adopt the automatic image processing of the equipment and only carry out the signal correction of each pixel were used for the analysis.

## 2. Respiratory frequency-adjusted dynamic image computation

The primary outcome of this study is list below.

1) XP is a three-dimensional structure, with thicker areas in front and behind having higher signal, and thinner areas having lower signal.

2) Whether the structure is drawn at the beginning of breathing, bronchus -> bronchioles -> alveoli, alveoli -> bronchioles -> bronchus according to the momentum during breathing.

3) Does the signal change along the wave of respiration and its depth?

4) Exclude artifact (rib artifact, lobular fissure artifact, diaphragm or mediastinum, hilum artifact)

**2–1. Segmentation of images at maximal inspiration and expiration.** The left lung field sets the apex of the lung (IP1, IP3), costotransverse angle (IP5), the intersection of the diaphragm and cardiac shadow (IP7), the intersection of the mediastinum and cardiac shadow (IP9), and the intersection of the mediastinum and aortic arch (IP11). In addition, IP2, IP4, IP6, IP8, IP10, and IP12 were set as auxiliary points for second-order Bezier curves, and the left lung field was contoured (Fig 1B). The right lung sets the apical lung (IP1, IP3), costotransverse angle (IP9), the intersection of the diaphragm and mediastinum (IP7), and a point on the mediastinum (IP5). The right lung field was also contoured by entering IP2, IP4, IP6, IP8, and IP10 as adjunct points to second-order Bezier curves (Fig 1A).

**2–2. Segmentation of images midway between expiration and inspiration.** Let the index of the image be t and let the xy coordinates of Pit be xit, and it, respectively ($i = 1, 2, \ldots, 12$). For all images, P5 was set to get y5t. y5max was y5 for maximal inspiratory images and y5min was y5 for maximal expiratory images. To achieve a maximum inspiratory image of 1 and a maximum expiratory image of 0, the percent change rt was calculated using the following formula:

$$r_t = \frac{y_{5 \cdot t} - y_{5 \cdot min}}{y_{5 \cdot max} - y_{5 \cdot min}}, \quad t = 1, 2, \ldots, N$$

The coordinates of IPi were set to ximax and yimax, and those of EPi were set to ximin and yimin; the lung field geometry Pit in each frame was calculated based on rt (Fig 2).

$$P_{i \cdot t} = \begin{pmatrix} x_{i \cdot t} \\ y_{i \cdot t} \end{pmatrix} = \begin{pmatrix} (1 - r_t) x_{i \cdot min} + r_t x_{i \cdot max} \\ (1 - r_t) y_{i \cdot min} + r_t y_{i \cdot max} \end{pmatrix}, \quad i = 1, 2, \ldots, 12$$

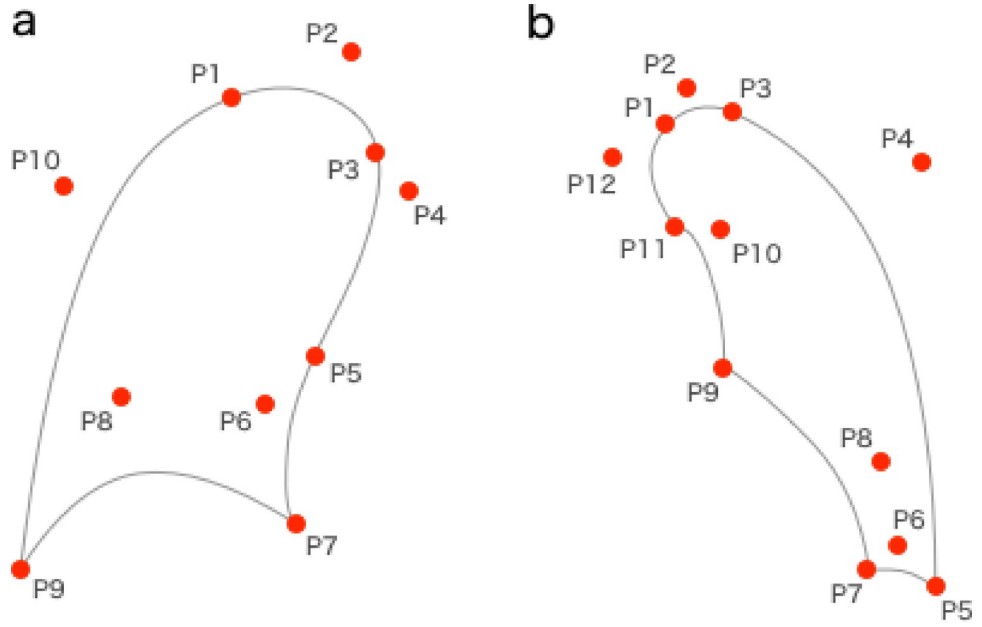

**Fig 1. Setting of both lung fields.**

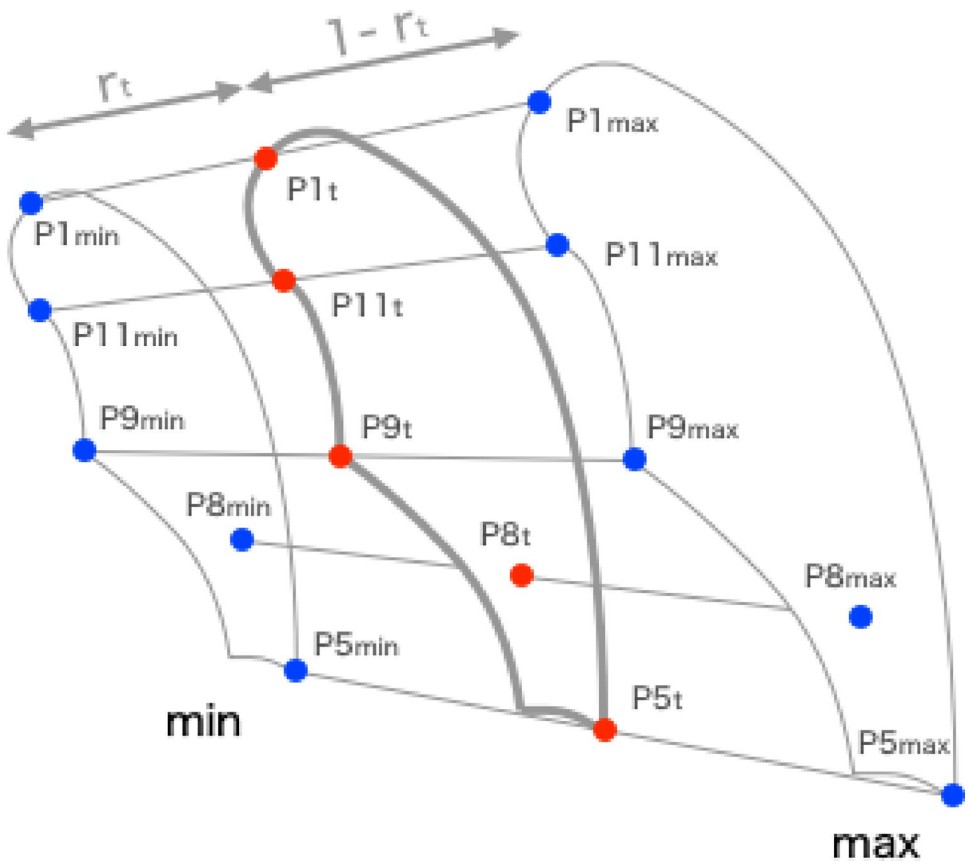

**Fig 2. Maximum inspiratory and expiratory images of the lungs.**

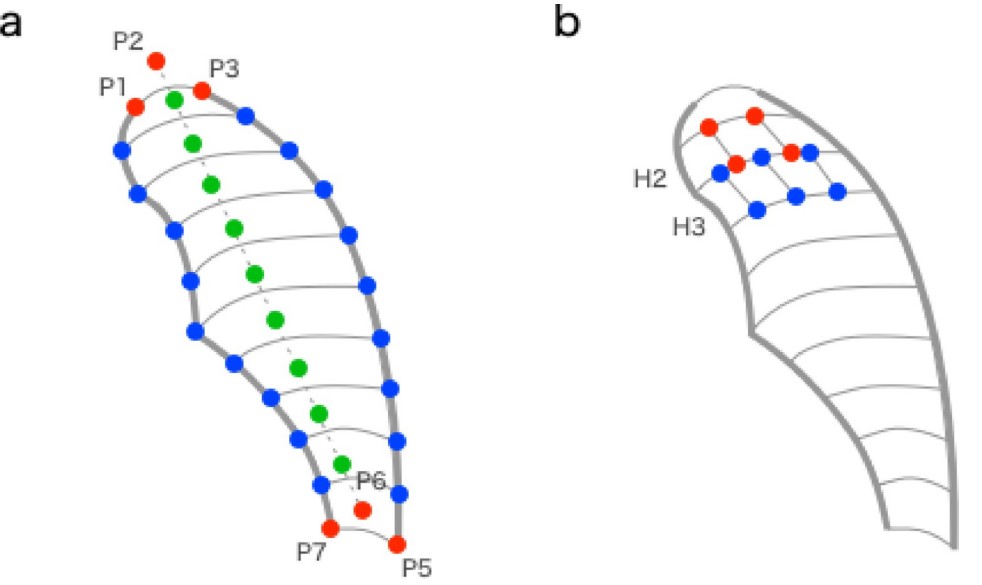

**Fig 3. Segmentation of the lungs.**

**2–3. Segmentation fragmentation.** The lung fields defined above were subdivided so that luminance changes in each region can be tracked. First, maximum expiration images (images with the smallest lung field) were used to obtain the segmentation numbers on the ordinate and abscissa. The curved P1P7 and P3P5 were defined as the vertical axis. Of these, the shorter one was divided by 4 pixels, and the number of divisions M was the number of divisions in the vertical axis direction. Then, the maximal expiratory images were segmented longitudinally in fractions M. In addition to the vertical axis, the supplementary line P2P6 was divided equally into M pieces, and the division point was used to draw a Bezier curve for M+1.

This Bezier curve divides the lung field into M pieces (Fig 3A). For each divided area, the short of the upper and lower Bezier curves was divided into 4-pixel lengths, and the divided number Nm was the number of divisions in the horizontal axis direction. The upper and lower Bezier curves were equally split into Nm pieces, and an Nm+1 line was drawn using the split point. The line further divided the longitudinally segmented region (Fig 3B). Segmentation was subdivided in all frames using the numbers of longitudinal segment M and transaxial segment Nm obtained using the method described above.

**2–4. Breath frequency-tuned dynamic image.** The average brightness values of each subdivided area were calculated, including the difference between the frames. Fourier transform was carried out for the changes in the luminance average values of each region, and the same frequency component as the respiration period was taken out. Inverse Fourier transform was performed, and it was converted into pixel values of the images of each phase. The values were used to display the signal values in color in the shape of the original lung field, and a respiratory frequency-adjusted dynamic image was created. The outline of the analysis is shown in the figure (Fig 4).

**3. Evaluation of the usefulness of the test.** The motion images calculated using the proposed technique were visually assessed by one radiologist and one respiratory physician. The evaluation method classified the results of uniformly changing signals in the whole left and right lungs as normal, while the conspicuous abnormal fluctuations were classified as serious. The intermediate was classified into five stages of almost normal, mild, and moderate disease by the radiologist and physician.

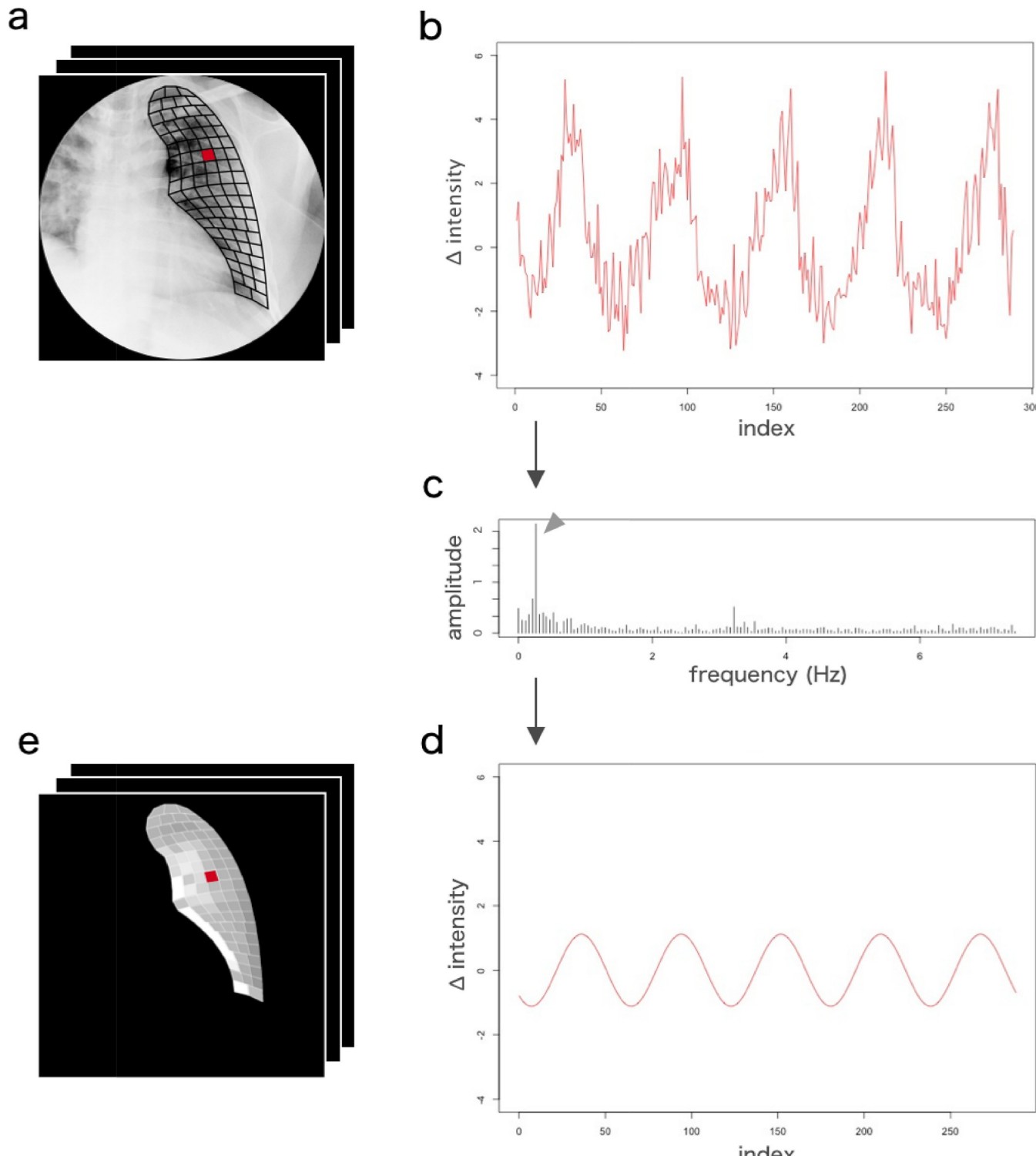

**Fig 4. Outline of analysis using breath frequency-tuned dynamic images.**

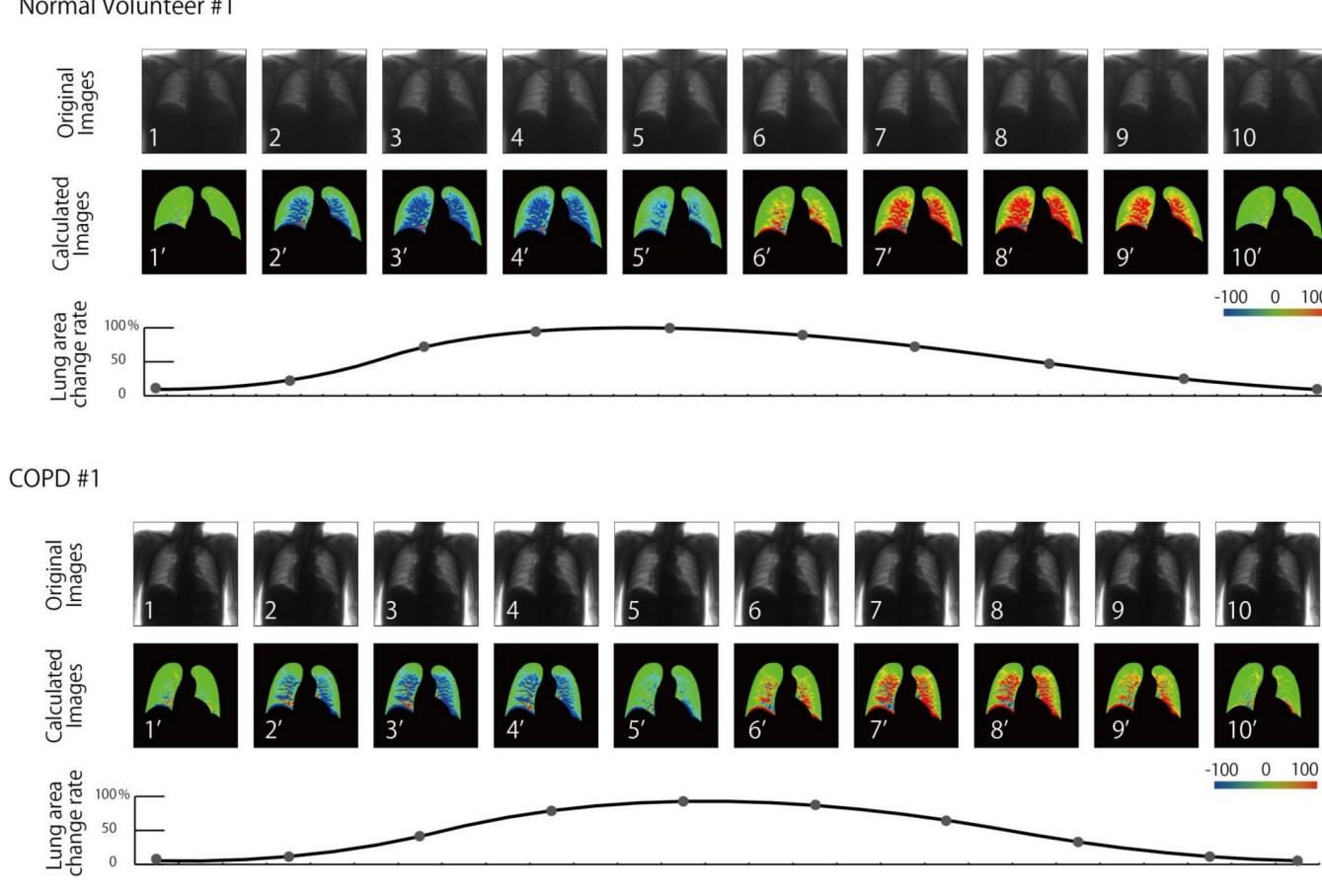

**Fig 5. Generated dynamic images of a normal volunteer (#1) and patient with COPD (#1).**

## Results

The mean total dose for all volunteers was 0.2 mGy. There was no major discrepancy in the detection of lung field geometry, and no diagnostic problems were noted by the radiologist and physician. Subsequent signal processing could also be performed without problems. The generated dynamic images are shown in Figs 5 and 6 and S1–S8 Movies. The expert evaluations are summarized in Table 1. As shown in Tables, differences were observed in patients classified as normal, GOLD1, and GOLD2. Although this study has a small case number, and the statistical processing was not carried out only by visual evaluation, there was a clear difference between the images. Each representative case was presented with serial images.

In all cases, the right and left lungs of the normal volunteers changed similarly in conjunction with the expiratory motion of the diaphragm, followed from the apex to near the diaphragm, and showed larger signal changes. No patchy patterns were observed in the signal changes in all lung fields, and no side-to-side differences were observed. Moreover, the signal rise and rib shape of the lung apex were strongly displayed in three normal volunteers, and the signal of the rib was lowered by fixing the motion of the rib in the thorax band. In one volunteer, the signal was slightly lower in the right upper lung field. One of the 15 patients did not differ significantly from normal patients, whereas in other cases, there was an incorrect movement in the signal changes in the lung fields. Three, seven, and four patients were judged as mild, moderate, and severe cases, respectively, with increasing severity as the rank of GOLD

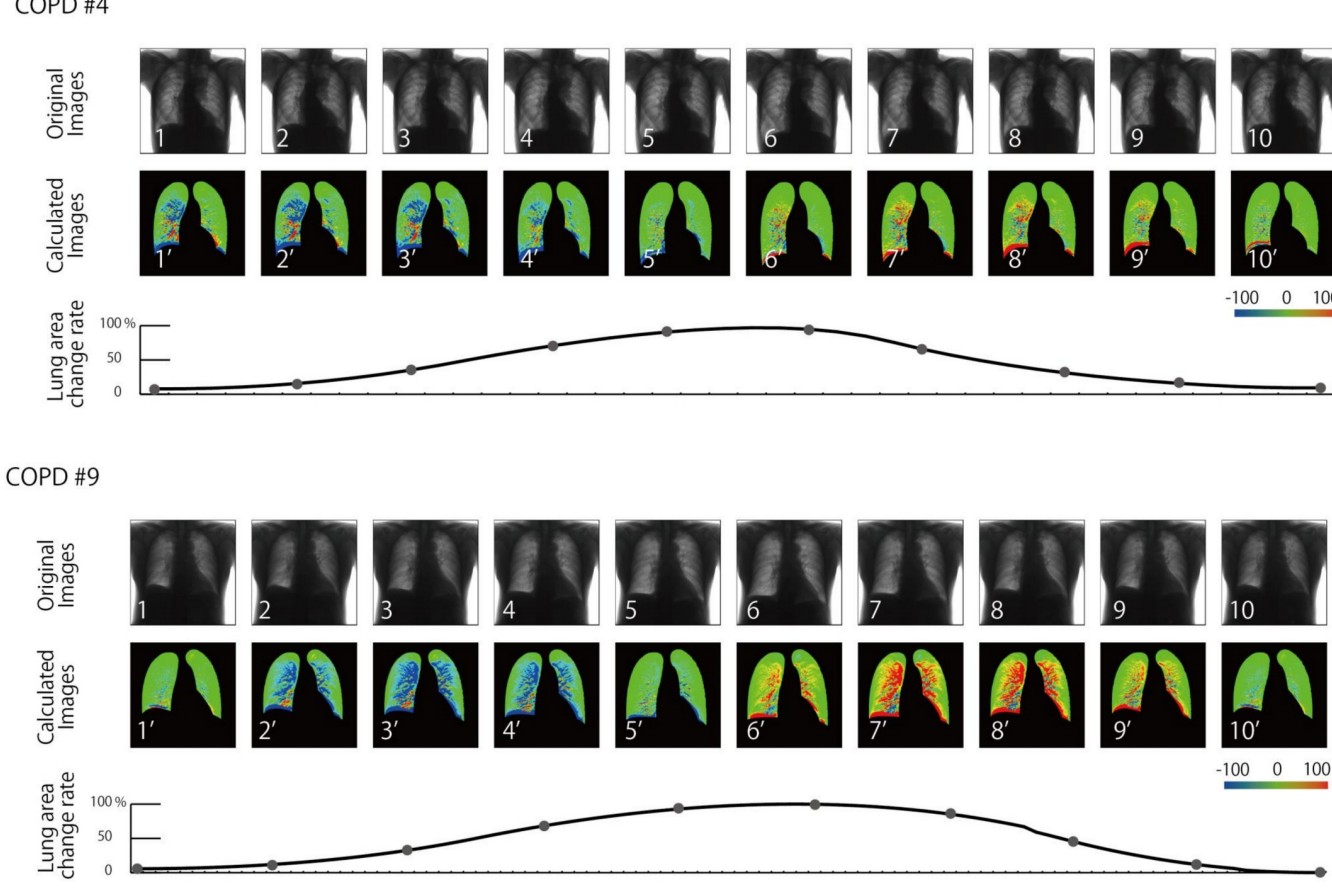

**Fig 6. Generated dynamic images of patients with COPD (#4 and #9).**

increased. Moreover, the motion of both right and left lungs was almost symmetric in five cases, while it became asymmetric in the remaining 10 cases.

## Discussion

Since the 1960s, studies have pointed out that there are various elements, such as respiratory elements, blood flow elements, and other noises, in radiography with XP [10]. However, to the best of our knowledge, there has been no approach to extract it with physiological logic and using engineering elements. By matching a band-pass filter to the respiratory frequency, respiratory frequency-adjusted images (Respiratory frequency-tunable imaging) could be acquired and delineated by removing the changes in the XP density of cardiovascular and other frequency elements. The extraction of respiratory physiology is a conscious component of respiration, but respiratory motion is usually performed with relatively regular cycles. Therefore, it may be reasonable to extract breathing images according to the frequency of breathing using the characteristics of periodic cycles when it is stable and periodic. There is also a mixture of various frequency elements besides breathing. In particular, the distribution components of blood flow could be removed by extracting only the frequency component of the respiratory cycle, although there was a mixture of cardiovascular frequency bands.

In this study, respiratory frequency-tuned images were successfully produced, and good images were acquired in a physiologically and technically organized fashion. The relative

**Table 1. Expert evaluations of each healthy volunteers and patients with COPD.**

| Participants | Age (years) | Sex | Percentage of FEV1 | GOLD stage of COPD | Image evaluation |
|---|---|---|---|---|---|
| Healthy volunteers | | | | | |
| 1 | 44 | Male | - | - | Normal |
| 2 | 49 | Male | - | - | Almost normal |
| 3 | 58 | Male | - | - | Normal |
| 4 | 38 | Female | - | - | Almost normal |
| 5 | 36 | Female | - | - | Normal |
| 6 | 25 | Male | - | - | Almost normal |
| 7 | 28 | Female | - | - | Normal |
| 8 | 26 | Male | - | - | Normal |
| 9 | 45 | Male | - | - | Normal |
| 10 | 33 | Male | - | - | Normal |
| Patients with COPD | | | | | |
| 1 | 74 | Male | 80.5 | 1 | Mild |
| 2 | 87 | Male | 100.4 | 1 | Moderate |
| 3 | 73 | Male | 100.3 | 1 | Moderate |
| 4 | 79 | Male | 71.9 | 2 | Severe |
| 5 | 75 | Male | 57.8 | 2 | Severe |
| 6 | 82 | Male | 93 | 1 | Mild |
| 7 | 82 | Male | 96.5 | 1 | Almost normal |
| 8 | 81 | Male | 79.4 | 2 | Moderate |
| 9 | 69 | Male | 75 | 2 | Moderate |
| 10 | 78 | Male | 84.5 | 1 | Moderate |
| 11 | 76 | Male | 53.8 | 2 | Severe |
| 12 | 72 | Male | 57.3 | 2 | Moderate |
| 13 | 83 | Male | 56.7 | 2 | Severe |
| 14 | 49 | Female | 84.3 | 1 | Almost normal |
| 15 | 66 | Male | 102.2 | 1 | Mild |

Abbreviations: COPD, chronic obstructive pulmonary disease; FEV1, forced expiratory volume in the first second; GOLD, global initiative for chronic obstructive lung disease

position of respiration could be traced by interlocking the lung field motion according to the respiration motion. Since the motion of the lung fields and the follow-up of the block in XP are strictly different stereologically, different signals were extracted even in positions like the lung hilum. However, the small block and scale of deformation changes were calculated, while a relatively good position was followed up as the whole lung field, so it was probable to obtain good images. The present normal shadow showed no signal changes in the apical region to bilateral lateral thorax, and strong signal changes were observed in the mediastinal diaphragmatic side. The gradation changes were relatively smooth. This was an anterior-posterior (AP) image-based image of XP, and the effect of the thickness of the AP lung structure was considered to be reflected as such. Thinner AP, apical, and bilateral peripheral zone thicknesses did not cause changes because of the thinner signal changes. More mediastinal or diaphragmatic AP lung structures became thicker, and signal changes became stronger accordingly. A relatively smooth increase in medial AP thickness may result in more uniform and larger changes. Differential images of the signal value changes were also generated because the changes in pixel values with respiration were imaged by taking the differences between adjacent frames. These changes seem to be physiologically consistent in terms of measuring respiration, and a

normal signal seems to be a condition in which the lung thickness in AP imaging and the motion of respiratory differential waves are well-reflected in the image.

With this image, there remains no tools or techniques that can visualize the physiological insights on how breathing itself functions partially in each part of the lung while following the shape changes in the lung fields. Moreover, stethoscopes can be listened to indirectly in real-time motion, but it is biased on a subjective judgment and cannot be visualized overall. Spirometry is the main pulmonary function test, but grasping the pulmonary resistance is the center, and it is different from grasping the condition of inherent lung respiration. However, it can be said that the real-time respiration condition is grasped in a precise pulmonary function test. Spirometry only grasped the dynamics of the whole lungs, and it could not visualize the condition of partial respiration. In the pulmonary function test by Kr or Xe, the lung function was depicted as an image, but it only reflected the results of the whole gas exchange, and it was difficult to partially evaluate the real-time inhalation and exhalation. Therefore, using physiological insights as a tool with simultaneous temporal and spatial resolution, we were able to show a new method for respiratory testing to prevent infectious diseases.

In this image, although there was a strong shadow in the lung apex of three volunteers, there was an improvement in the fitting of the thorax breathing band. It was assumed that the motion of the pectoralis major muscle was reflected in the element because thoracic breathing became more dominant than abdominal breathing in the youth. Compared to the above normal findings, the signals in patients with COPD generally differed, causing partial signal reduction and becoming a mosaic signal. It was also accompanied by a reversed phase signal during respiration in several ranges. These were considered to be due to fibrosis and emphysematous changes. Due to the inability to achieve a homogeneous lung stretch and constriction and partial traction changes, the signal in the opposite direction became heterogeneous. In addition, as COPD became more severe, there was a reduction in heterogeneity and global signals. These changes may reflect the lung-wide heterogeneity, segmental changes, and global hypointense changes due to the progression of lung fibrosis and emphysematous changes caused by COPD.

In this image, a shadow that seems to be some artifact is shown. Those associated with rib migration and deformation and those associated with the individual movements of the lung lobes were considered to be artifacts due to signal processing. These artifacts must be investigated in the future. Breathing has a conscious component, but it is practiced at rest in relatively regular cycles. Extracting signals according to the frequency of breathing is considered practical when breathing is stable [11, 12].

In this study, the parameters of X-ray exposure in the fluoroscopy equipment were set using the automatic setting function. Since the images were taken using existing fluoroscopy equipment settings, it was considered that there was a room for a significant reduction in the exposure dose by implementing an appropriate pulsed radiation. The optimum pulse rate in breathing at that time should be considered by simulation. Frame rates and allowable noise levels also seemed to be important to consider. It is also important to examine whether there are adjuncts, such as photographing posture and bust band and resting breathing guidance. It should be noted that pregnant women and young people should be avoided because of x-ray exposure. To use the relative positions and sizes of the block along the respiratory function, a relatively good image was obtained. In the case of irregular physiological changes, the image was easy to be produced as an artifact, which may require software and photographing method adjustments. There are many hospitals with fluoroscopy devices, and new facility investment are not needed. At present, when spirometry is not recommended, this technique will be a useful test in the future.

The signal was generated in accordance with the periodic component of respiration of XP, thus removing the periodic component of blood flow, the frequency components of other organs, and the synthetic wave component. The evaluation of each frequency is different from the focus of this study and may be necessary in other work.

The imbalance of ventilation is of course an issue, and together with the dynamics of pulmonary blood flow, which is evaluated at the same time, is likely to be reflected in cardiopulmonary function and exercise capacity, but this is not the focus of this study.

The relevance of this finding to the way patients move is an issue for the future. In this study, we first took data from patients with COPD. Although there are other physiological factors in COPD that are associated with ventilatory imbalance, it is future work to reflect these factors in the testing methods presented in this paper.

The method presented in this study can be performed with all angiography and fluoroscopy equipment, provided that continuous fluoroscopic images can be stored, images can be taken under fixed fluoroscopic conditions, and images can be output in DICOM, respectively. In addition, images can be output in as little as 5 minutes. Although some issues, such as rib delineation, may cause variability in interpretation of results, this method seems to be one major indicator. The usefulness of this proposed technique could be an alternative to spirometry, but its clinical usefulness in many diseases needs to be further examined. We plan to increase and confirm the clinical cases in the future.

There were some limitations. The visual evaluations were ultimately judged jointly by the radiologist and the respiratory physician. In this study, the FEV1 was used to clinically perform GOLD classification to determine the grade of COPD. Based on these results, we are comparing the results with the images. The correlation is not calculated due to the small sample size and the fact that this is not the focus of this study.

## Conclusion

Existing XP cine imaging was used to extract frequency-tunable imaging. Using the physiological breathing frequency of a normal patient, a relatively smooth image, which depended on the differential wave of breathing difference and thickness, was acquired. With the worsening of COPD, there were native respiratory differential wave and inverted signals, with a global signal reduction and heterogeneous signal change.

In this study, examination using serial fluoroscopic images was proposed as an alternative to spirometry. The imaging is simple and can be performed conveniently without requiring forced expiration.

Compared to spirometry, this examination can identify the abnormal regions on the images, and it does not require maximum effort respiration. Therefore, it is possible to perform a stable examination because the patient's physical condition and the ability of laboratory technicians on the day are less affected. This can also be used as a countermeasure in examining patients with infectious diseases.

## Supporting information

**S1 Movie. The generated dynamic movie 1.**
(MP4)

**S2 Movie. The generated dynamic movie 2.**
(MP4)

**S3 Movie. The generated dynamic movie 3.**
(MP4)

**S4 Movie. The generated dynamic movie 4.**
(MP4)

**S5 Movie. The generated dynamic movie 5.**
(MP4)

**S6 Movie. The generated dynamic movie 6.**
(MP4)

**S7 Movie. The generated dynamic movie 7.**
(MP4)

**S8 Movie. The generated dynamic movie 8.**
(MP4)

**S1 Dataset. The dataset of this manuscript.**
(ZIP)

## Author Contributions

**Conceptualization:** Takehiko Abe.

**Data curation:** Norifumi Yoshida, Masanao Nakashima, Atsushi Nagai.

**Formal analysis:** Takehiko Abe, Masanao Nakashima.

**Investigation:** Takehiko Abe.

**Methodology:** Takehiko Abe, Norifumi Yoshida.

**Software:** Takehiko Abe, Norifumi Yoshida.

**Visualization:** Norifumi Yoshida.

**Writing – original draft:** Takehiko Abe, Norifumi Yoshida, Tetsuo Shimada.

**Writing – review & editing:** Takehiko Abe, Norifumi Yoshida, Tetsuo Shimada, Masanao Nakashima, Atsushi Nagai.

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
