## [Decision Letter · Decision Letter 0]

9 Aug 2022

PONE-D-21-37867A proposed new examination method using chest X-ray cine imaging: Respiratory frequency-tunable imaging for lung function visualization by Fourier analysis considering various respiratory diseases, including COVID-19PLOS ONE

Dear Dr. Abe,

Thank you for submitting your manuscript to PLOS ONE. After careful consideration, we feel that it has merit but does not fully meet PLOS ONE’s publication criteria as it currently stands. Therefore, we invite you to submit a revised version of the manuscript that addresses the points raised during the review process. Please submit your revised manuscript by Sep 23 2022 11:59PM. If you will need more time than this to complete your revisions, please reply to this message or contact the journal office at plosone@plos.org. Please include the following items when submitting your revised manuscript:A rebuttal letter that responds to each point raised by the academic editor and reviewer(s). You should upload this letter as a separate file labeled 'Response to Reviewers'.A marked-up copy of your manuscript that highlights changes made to the original version. You should upload this as a separate file labeled 'Revised Manuscript with Track Changes'.An unmarked version of your revised paper without tracked changes. You should upload this as a separate file labeled 'Manuscript'.

We look forward to receiving your revised manuscript.

Kind regards,

Aleksandra Barac

Academic Editor

PLOS ONE

Journal Requirements:

3. PLOS requires an ORCID iD for the corresponding author in Editorial Manager on papers submitted after December 6th, 2016. Please ensure that you have an ORCID iD and that it is validated in Editorial Manager. To do this, go to ‘Update my Information’ (in the upper left-hand corner of the main menu), and click on the Fetch/Validate link next to the ORCID field. This will take you to the ORCID site and allow you to create a new iD or authenticate a pre-existing iD in Editorial Manager. Please see the following video for instructions on linking an ORCID iD to your Editorial Manager account: https://www.youtube.com/watch?v=_xcclfuvtxQ.

Reviewers' comments:

Reviewer's Responses to Questions

**Comments to the Author**

1. Is the manuscript technically sound, and do the data support the conclusions?

Reviewer #1: Yes

Reviewer #2: Yes

2. Has the statistical analysis been performed appropriately and rigorously? 

Reviewer #1: Yes

Reviewer #2: No

3. Have the authors made all data underlying the findings in their manuscript fully available?

Reviewer #1: Yes

Reviewer #2: Yes

4. Is the manuscript presented in an intelligible fashion and written in standard English?

Reviewer #1: Yes

Reviewer #2: Yes

5. Review Comments to the Author

Reviewer #1: Thank you for your valuable work.

The authors presented an interesting research work of respiratory frequency-tunable imaging aiming to apply in lung function visualization as a suggested new method of investigation. The new examination method, as the authors explained, will be a fruitful alternative to the traditional pulmonary function testing in situations where there is a fear of infection transmission during a pandemic like the current SARS COV2, also, in cases when the patient is unable to do the forced maneuver due to weakness or a probable contraindication.

I appreciated this idea especially the manuscript is also well presented, and the authors followed a good scientific approach.

Title:

• Informative, reflects the paper’s content.

Abstract:

• Well written, well structured, informative.

Introduction:

• Adequate background information.

• References are adequate.

• Stated the specific study objectives.

• Writing is clear and concise.

Methods:

• The outcome variables are well described.

• The measurement procedures as well as statistical methods chosen are appropriate.

• The writing is clear.

Results:

• The data are well presented, well written.

Discussion:

• Well presented, good writing.

• Adequate references and comparative discussion.

References:

• Adequate and appropriately cited.

• The authors adopted recent references.

Tables: are clear.

Figures: are clear.

Videos are reasonable and of acceptable quality.

Reviewer #2: Τhe authors in this interesting study aimed to assess respiratory function using continuous fluoroscopic images during breathing. Their method presents some advantages as less exposure than scintigraphy and CT, and no need for contrast media. The idea is intriguing – although the concept not entirely novel. They present nicely their technique and I see from the literature that they have experience on this field. However, I have some concerns both for the method and for the paper itself. The authors have not provided evidence that the method is comparable in terms of physiology or clinical usefulness with other established tests. The paper is rather a technical report and it could be presented better in this form.

In more detail

1. Title. The title is too long

2. Abstract. It does not represent well the paper, especially in the section of results. Conclusions are also not well supported by the results and the section for an abstract is long.

3. Purpose. The purpose is rather confusing as it is expressed in the text.

4. Methods. The authors should define better their primary outcome, independent and dependent variables and method of their assessment.

5. Results. Based on the aim of the study, one expects that that there will be a correlation between the results of the presented method and spirometry. However, I do not see such a test but a brief report for some cases. The authors should either quantify the results of continuous fluoroscopy so they can correlate them with spirometry. Either wise if they cannot quantify the results of the test in a simple variable they should pick another hypothesis i.e. continuous fluoroscopy correlates with exercise capacity i.e. 6mwt, BODE etc

6. The authors present no evaluation of agreement between observers.

7. Discussion.

I think that there are some main points that have to be depicted in this study. Feasibility, simplicity, variability of the interpretation of the results, physiologic meaning and clinical utility. I believe that authors should attempt to provide answers to the above in a clear way.

8. To my view a key point in discussion is in the paragraph where the authors report that the signals in patients with COPD generally differed, causing partial signal reduction and becoming a mosaic signal. As COPD became more severe, there was a reduction in heterogeneity and global signals which may reflect the lung-wide heterogeneity, segmental changes, and global hypointense changes due to the progression of lung fibrosis and emphysematous changes caused by COPD. I agree but the authors should provide more insight for the production of the signal. They use a technique that reflects regional ventilation differences which may not correlate so well with dynamic flows (i.e. FEV1). There are other physiologic elements in COPD which might be related to ventilation inequalities such as IC, FRC, iPEEP, Compliance. I suggest that, at the end, ventilation inequalities could be better reflected in exercise capacity. In this respect I suggest to correlated the technique with a variable that reflects exercise capacity.

6. PLOS authors have the option to publish the peer review history of their article (what does this mean?). If published, this will include your full peer review and any attached files.

Reviewer #1: **Yes: **Hesham Atef AbdelHalim

Reviewer #2: **Yes: **DEMOSTHENES MAKRIS

---

## [Author Response · Author response to Decision Letter 0]

13 Oct 2022

Answer for Reviewer 1

Thank you for your valuable work.

The authors presented an interesting research work of respiratory frequency-tunable imaging aiming to apply in lung function

visualization as a suggested new method of investigation. The new examination method, as the authors explained, will be a fruitful

alternative to the traditional pulmonary function testing in situations where there is a fear of infection transmission during a pandemic

like the current SARS COV2, also, in cases when the patient is unable to do the forced maneuver due to weakness or a probable

contraindication.

I appreciated this idea especially the manuscript is also well presented, and the authors followed a good scientific approach.

Title:

• Informative, reflects the paper’s content.

Abstract:

• Well written, well structured, informative.

Introduction:

• Adequate background information.

• References are adequate.

• Stated the specific study objectives.

• Writing is clear and concise.

Methods:

• The outcome variables are well described.

• The measurement procedures as well as statistical methods chosen are appropriate.

• The writing is clear.

Results:

• The data are well presented, well written.

Discussion:

• Well presented, good writing.

• Adequate references and comparative discussion.

References:

• Adequate and appropriately cited.

• The authors adopted recent references.

Tables: are clear.

Figures: are clear.

Videos are reasonable and of acceptable quality.

Thank you very much for your good comments.

 

Answer for Reviewer 2

The authors in this interesting study aimed to assess respiratory function using continuous fluoroscopic images during breathing. Their method presents some advantages as less exposure than scintigraphy and CT, and no need for contrast media. The idea is intriguing – although the concept not entirely novel. They present nicely their technique and I see from the literature that they have experience on this field. However, I have some concerns both for the method and for the paper itself. The authors have not provided evidence that the method is comparable in terms of physiology or clinical usefulness with other established tests. The paper is rather a technical report and it could be presented better in this form.

Thank you very much for your comments. We will respond to your individual points as follows.

1. Title. The title is too long

 Thank you very much for your comments. Based on your remarks, we will change the title to the following.

 Respiratory frequency-tunable dynamic imaging for lung function: New exam method using chest X-ray cine imaging considering various respiratory diseases

2. Abstract. It does not represent well the paper, especially in the section of results. Conclusions are also not well supported by the results and the section for an abstract is long.

Thank you very much for your comments. Based on your remarks, we have revised the Results in the Abstract. The revisions are shown below.

The mean total dose for all volunteers was 0.2 mGy. There was no major discrepancy in the detection of lung field geometry, and no diagnostic problems were noted by the radiologist and physician.

3. Purpose. The purpose is rather confusing as it is expressed in the text.

Thank you very much for your suggestion. We will change the first section to Introduction, and the Purpose will also be listed in the Introduction.

4. Methods. The authors should define better their primary outcome, independent and dependent variables and method of their assessment.

Thank you very much for your suggestion. The following text has been added to the Materials and Methods section as the primary outcome.

The primary outcome of this study is listed below.

1) XP is a three-dimensional structure, with thicker areas in front and behind having higher signal, and thinner areas having lower signal.

2) Whether the structure is drawn at the beginning of breathing, bronchus -> bronchioles -> alveoli, alveoli -> bronchioles -> bronchus according to the momentum during breathing.

3) Does the signal change along the wave of respiration and its depth?

4) Exclude artifact (rib artifact, lobular fissure artifact, diaphragm or mediastinum, hilum artifact)

5. Results. Based on the aim of the study, one expects that that there will be a correlation between the results of the presented method and spirometry. However, I do not see such a test but a brief report for some cases. The authors should either quantify the results of continuous fluoroscopy so they can correlate them with spirometry. Either wise if they cannot quantify the results of the test in a simple variable they should pick another hypothesis i.e. continuous fluoroscopy correlates with exercise capacity i.e. 6mwt, BODE etc

Thank you very much for your attention. The following sentences have been added to the Limitation in Discussion section.

In this study, the FEV1 was used to clinically perform GOLD classification to determine the grade of COPD. Based on these results, we are comparing the results with the images. The correlation is not calculated due to the small sample size and the fact that this is not the focus of this study.

6. The authors present no evaluation of agreement between observers.

Since the judgment in this study was made jointly by radiologists and respiratory physicians, individual judgment was not made. Therefore, the following text has been added to the Limitation.

The visual evaluations were ultimately judged jointly by the radiologist and the respiratory physician.

7. Discussion.

I think that there are some main points that have to be depicted in this study. Feasibility, simplicity, variability of the interpretation of the results, physiologic meaning and clinical utility. I believe that authors should attempt to provide answers to the above in a clear way.

Thank you for your suggestion. Based on your suggestion, we have added the following sentences to the Discussion.

The method presented in this study can be performed with all angiography and fluoroscopy equipment, provided that continuous fluoroscopic images can be stored, images can be taken under fixed fluoroscopic conditions, and images can be output in DICOM, respectively. In addition, images can be output in as little as 5 minutes. Although some issues, such as rib delineation, may cause variability in interpretation of results, this method seems to be one major indicator.

8. To my view a key point in discussion is in the paragraph where the authors report that the signals in patients with COPD generally differed, causing partial signal reduction and becoming a mosaic signal. As COPD became more severe, there was a reduction in heterogeneity and global signals which may reflect the lung-wide heterogeneity, segmental changes, and global hypointense changes due to the progression of lung fibrosis and emphysematous changes caused by COPD. I agree but the authors should provide more insight for the production of the signal. They use a technique that reflects regional ventilation differences which may not correlate so well with dynamic flows (i.e. FEV1). There are other physiologic elements in COPD which might be related to ventilation inequalities such as IC, FRC, iPEEP, Compliance. I suggest that, at the end, ventilation inequalities could be better reflected in exercise capacity. In this respect I suggest to correlated the technique with a variable that reflects exercise capacity.

Thank you for your suggestion. Based on your suggestion, we have added the following sentences to the Discussion.

The signal was generated in accordance with the periodic component of respiration of XP, thus removing the periodic component of blood flow, the frequency components of other organs, and the synthetic wave component. The evaluation of each frequency is different from the focus of this study and may be necessary in other work.

The imbalance of ventilation is of course an issue, and together with the dynamics of pulmonary blood flow, which is evaluated at the same time, is likely to be reflected in cardiopulmonary function and exercise capacity, but this is not the focus of this study.

The relevance of this finding to the way patients move is an issue for the future. In this study, we first took data from patients with COPD. Although there are other physiological factors in COPD that are associated with ventilatory imbalance, it is future work to reflect these factors in the testing methods presented in this paper.

---

## [Editor Report · Decision Letter 1]

17 Oct 2022

Respiratory frequency-tunable dynamic imaging for lung function: New exam method using chest X-ray cine imaging considering various respiratory diseases

PONE-D-21-37867R1

Dear Dr. Abe,

We’re pleased to inform you that your manuscript has been judged scientifically suitable for publication and will be formally accepted for publication once it meets all outstanding technical requirements.

Kind regards,

Aleksandra Barac

Academic Editor

PLOS ONE

---

## [Editor Report · Acceptance letter]

27 Oct 2022

PONE-D-21-37867R1 

Respiratory frequency-tunable dynamic imaging for lung function: New exam method using chest X-ray cine imaging considering various respiratory diseases 

Dear Dr. Abe:

I'm pleased to inform you that your manuscript has been deemed suitable for publication in PLOS ONE. Congratulations! Your manuscript is now with our production department. 

Kind regards, 

on behalf of

Dr. Aleksandra Barac 

Academic Editor

PLOS ONE